# Generalised anxiety disorder and hospital admissions: findings from a large, population cohort study

Olivia Remes, Nicholas Wainwright, Paul Surtees, Louise Lafortune, Kay-Tee Khaw, Carol Brayne

Department of Public Health and Primary Care, University of Cambridge, Cambridge, UK

**Correspondence to**
Olivia Remes;
or260@medschl.cam.ac.uk

## ABSTRACT

**Objective** Generalised anxiety disorder (GAD) is the most common anxiety disorder in the general population and has been associated with high economic and human burden. However, it has been neglected in the health services literature. The objective of this study is to assess whether GAD leads to hospital admissions using data from the European Prospective Investigation of Cancer-Norfolk. Other aims include determining whether early-onset or late-onset forms of the disorder, episode chronicity and frequency, and comorbidity with major depressive disorder (MDD) contribute to hospital admissions.

**Design** Large, population study.

**Setting** UK population-based cohort.

**Participants** 30 445 British participants were recruited through general practice registers in England. Of these, 20 919 completed a structured psychosocial questionnaire used to identify presence of GAD. Anxiety was assessed in 1996–2000, and health service use was captured between 1999/2000 and 2009 through record linkage with large, administrative health databases. 17 939 participants had complete data on covariates.

**Main outcome measure** Past-year GAD defined according to the Diagnostic and Statistical Manual of Mental Disorders, fourth edition.

**Results** In this study, there were 2.2% (393/17 939) of respondents with GAD. Anxiety was not independently associated with hospital admissions (incidence rate ratio (IRR)=1.04, 95% CI 0.90 to 1.20) over 9 years. However, those whose anxiety was comorbid with depression showed a statistically significantly increased risk for hospital admissions (IRR=1.23, 95% CI 1.02 to 1.49).

**Conclusion** People with GAD and MDD comorbidity were at an increased risk for hospital admissions. Clinicians should consider that meeting criteria for a pure or individual disorder at one point in time, such as past-year GAD, does not necessarily predict deleterious health outcomes; rather different forms of the disorder, such as comorbid cases, might be of greater importance.

### Strengths and limitations of this study

► We used a large sample of British people recruited from the general population, and controlled for a number of relevant confounders, such as medical history and risk behaviours.
► We used a structured, self-reported questionnaire to determine whether participants met criteria for past-year GAD according to the DSM-IV.
► We examined health service use through record linkage with large, administrative health databases.
► Respondents were less deprived and healthier than individuals living in other regions of England; therefore, our results may not be generalisable to people living in extremely deprived circumstances.

## INTRODUCTION

Anxiety disorders[1] are the most common class of psychiatric disorders in the general population. The Global Burden of Disease study[2] estimated that anxiety disorders contribute to roughly 26.8 million disability-adjusted life-years, and their annual direct cost is $42.3 billion.[3] Generalised anxiety disorder (GAD) is defined by pervasive worry, and a number of additional symptoms, such as restlessness, muscle tension, and concentration difficulties. It is a prevalent and disabling condition in adults and can lead to impairment in social and occupational functioning.[4] GAD is associated with poor quality of life and risk of suicide.[5–8] Out of the anxiety disorders, this condition has been found to be the most impairing[5], and has been linked to high cortisol levels.[9] Although there is effective treatment for GAD, a minority of those affected receive any treatment.[8] This is because anxiety disorders are frequently under-recognised and mismanaged by clinicians in primary care, which is often the first point of contact for those with mental health problems.[10]

Although detection of anxiety in clinical settings is poor[11 12] and the presence of undiagnosed mental health problems can contribute to further emotional distress in patients down the line,[12] it could be that disorders such as GAD represent more than just psychological or worry-related symptoms. It may be that anxiety symptoms are masking underlying poor physical health or could be

an early warning signal for future health problems that are not yet detectable by standard medical tests. Such problems cannot be simply resolved through psychological therapies or psychotropic medication.

Anxiety has been linked to hypothalamic–pituitary–adrenal axis dysregulation and inflammation, and this can lead to poor health.[9] A recent study of hospitalised patients[13] also showed that people with anxiety disorders had more comorbid physical conditions compared with people without anxiety disorders. Conversely, anxiety could also represent a response to underlying medical illness, and physical illness can exacerbate anxiety; the possibility of a bidirectional relationship between anxiety and physical health should not be excluded.[14 15] Compelling evidence from prospective studies, however, has shown that anxiety can indeed increase the risk of serious chronic conditions, such as cancer[16] and coronary heart disease (CHD).[17]

When investigating the links between mental disorders and health outcomes, early-onset or late-onset forms of anxiety disorders, as well as psychiatric comorbidity, should be considered. A large study[17] of over >1 million Swedish men followed for over >20 years showed that early-onset forms of mental disorders led to increased risk of incident CHD. Anxiety disorders, such as GAD, are also frequently comorbid with major depressive disorder (MDD),[18] and psychiatric comorbidity has been associated with poorer quality of life, worse prognosis and higher use of health services for mental health problems than pure forms of the disorder.[19–21] Therefore, identifying clinical aspects, such as early-onset or late-onset forms of the condition, episode chronicity and frequency, and comorbidity with MDD, can lead to better clinical management and insight into groups at high risk for health service use.[22]

GAD is one of the most common anxiety disorders in the general population[23] and the primary care setting,[24] and has been associated with high economic and human burden. However, it has been largely neglected in the health services literature, with the exception of some studies showing GAD to contribute to higher use of primary care services in primary care samples.[24–28] Clinical samples, however, have the potential for self-selection bias. Further research is needed to determine whether GAD leads to hospital admissions.

The objective of this study is to assess the association between GAD and hospital admissions in a longitudinal, population cohort of >18 000 British individuals followed for 9 years. The aim is also to determine whether early-onset or late-onset forms of the disorder, episode frequency and chronicity, and comorbidity with MDD contribute to health service use.

## METHODS
### Study population
The study population was drawn from the European Prospective Investigation of Cancer-Norfolk (EPIC-Norfolk), a longitudinal, cohort study.[29 30] Between 1993 and 1997, a total of 30 445 participants over the age of 40, living in Norwich and the surrounding towns and rural areas were identified through general practice registers and recruited. At baseline, respondents were sent a health questionnaire, which captured information on sociodemographics, medical history, and risk behaviours. Between 1993 and 1999/2000, participants completed self-reported postal questionnaires provided they remained on the study's mailing list, had a valid mailing address, and were still alive. Between 1996 and 1999/2000, respondents completed a psychosocial questionnaire, the Health and Life Experiences Questionnaire (HLEQ),[29] which was used to capture information on psychiatric disorders and disability. Record linkage with administrative health databases using a unique identifier was used to ascertain hospital admissions until 2009.

Respondents who completed a baseline health questionnaire between 1993 and 1997 were eligible for study inclusion; those who completed a psychosocial questionnaire between 1996 and 1999/2000 were eligible for inclusion in the analysis.

### Assessment of GAD
Past-year GAD was defined according to the Diagnostic and Statistical Manual of Mental Disorders, fourth edition (DSM-IV), and derived using the HLEQ. The psychosocial questionnaire captured the onset and offset timings of episodes of anxiety disorder.[31] Past-year GAD consisted of at least one episode that had offset within 12 months of administration of the HLEQ. DSM-IV GAD was identified if respondents endorsed uncontrollable, excessive worry for 6 months or longer that contributed to life interference and help-seeking. At least three of the following symptoms needed to also have been present: restlessness, irritability, muscle tension, fatigue, trouble concentrating because of worry, mind going blank, trouble falling asleep, trouble staying asleep and feeling keyed up or on edge.

### Assessment of covariates
Potential confounders (based on the literature) included sociodemographics (age, sex, education, marital status, social class, employment), prevalent physical diseases, disability, MDD and risk behaviours (alcohol use, smoking, physical activity). The categorisation of the confounders included in the models took account of cell size and was also done in accordance with previous research.[31–37] Age was examined as a categorical variable and then divided into 10-year bands. Sex was categorised into male versus female; marital status was divided into married, single (or never married) and others (widowed, divorced, separated); educational level into high (vocational or formal qualifications at the A-level or O-level or degree-level qualifications) versus low (no formal qualifications). The social class variable was created based on the Computer-Assisted Standard Occupational Coding[37] and categorised into the following groups: I (professionals), II (managerial and technical occupations), III

non-manual and III manual (skilled workers), IV (partly skilled workers) and V (unskilled manual workers). For men, social class was assigned based on their own occupation. Unemployed men were assigned the social class of their partner, while retired men were assigned social class based on their last emploment. Unemployed men without partners were unclassified. Social class for women was coded using their partners'. However, if the woman had no partner, or the partner's social class was unclassified or missing, social class was based on (the woman's) own occupation. An unemployed woman without a partner was coded as unclassified. The final categorisation of social class was manual (skilled manual, partly skilled and unskilled) versus non-manual (professionals, managerial and technical, and skilled non-manual). Employment was categorized into yes versus no.

Behaviour risk factor measures included alcohol intake (units of alcohol/week), smoking status (current, former, non-smoker) and physical activity (inactive, moderately inactive, moderately active, active). Presence of past-year MDD (yes/no) defined according to the DSM-IV was also examined.[38]

Individual-level health status measured the presence of major prevalent physical diseases associated with anxiety ([self] reporting at least one of the following physician-diagnosed conditions: asthma, bronchitis, allergies, hay fever, stroke, heart attack, cancer, diabetes, thyroid conditions, arthritis).[39] Disability was measured using the physical component summary (PCS) score of the Medical Outcomes Study 36-Item Short Form, a valid self-assessment instrument. Higher scores correspond to better health while lower scores indicate poorer health.[40]

The variables described were potential confounders and selected based on previous research.[41 42]

### Hospital service use

All analyses are based on hospitalisations. Primary care service use was not captured in this study.

Frequency of hospitalisation between 1999/2000 and 2009 was determined using the East Norfolk Primary Health Care trust databases, which capture publicly funded healthcare activity data and administrative information from facilities, such as hospitals.

England is under a publicly funded healthcare system (the National Health Service), free at the point of delivery; therefore, we expect factors, such as access to health insurance or personal income, to have a minimal impact on the care that is obtained by study participants. The databases used in this study are likely to capture most hospital admissions from the population, as private sector provision is minimal. This means that admissions data in our study can be considered complete for the ascertainment of hospital/health service use, and the likelihood of bias minimal. To access hospital services in the UK, a referral is needed from the primary care practitioner, who acts as a gate-keeper to secondary care.

The East Norfolk Primary Health Care databases were linked to the EPIC-Norfolk cohort using participants'

unique National Health Service numbers, which allow complete record linkage across settings and time.

Vital status for participants was determined through record linkage with the United Kingdom Office of National Statistics. Since vital status was available for all respondents, we were able to exclude those who died before their health service use was ascertained.

### Statistical analysis

First, sociodemographics were compared by GAD status—Pearson's $\chi^2$ test was used to determine whether differences were statistically significant for categorical variables. Second, the mean number of hospital admissions was determined for each characteristic/covariate—the Kruskal-Wallis test was used to determine statistical significance for categorical covariates with three or more categories, while the Wilcoxon rank-sum test was used for dichotomous covariates.

The number of hospital admissions was skewed and the variance was much larger than the mean. The models examining the relation between different forms of GAD and hospital service use (number of hospital admissions) were based on zero-inflated negative binomial regression. Three models were fitted for hospital admissions with progressive adjustment of confounders: model A adjusted for sociodemographics (age, sex, education, marital status, social class, employment), physical conditions and disability; model B further accounted for past-year MDD (assessed at the same time as past-year GAD); and model C further controlled for physical activity, alcohol and smoking.

Finally, we determined whether the risk for hospitalisation was higher among those with (1) ≥3 episodes of lifetime GAD (vs those with <3 episodes or no GAD), (2) episodes that lasted on average ≥6 months (vs those with <6 months or no GAD), (3) age of onset at 30 years or younger (vs people with age at onset >30 years or no GAD) and (4) psychiatric comorbidity with MDD (vs no GAD-MDD comorbidity). Two-sided statistical tests for the parameter estimates were conducted, and a P value of <0.05 was considered statistically significant. Analyses were implemented in SAS V.9.3.

To arrive at the study size, we did the following: of the 30 445 who completed the baseline HLQ, we retained those participants who completed the psychosocial HLEQ (20 919), and of these, we kept respondents with complete data on covariates (17 939) (figure 1).

### Patient involvement

There were no patients involved in the development of the research question and outcome measures, the design of the study or the recruitment to and conduct of the study.

### RESULTS

Of the 30 445 people who consented to participate in EPIC-Norfolk and completed the HLQ at baseline, 20 919

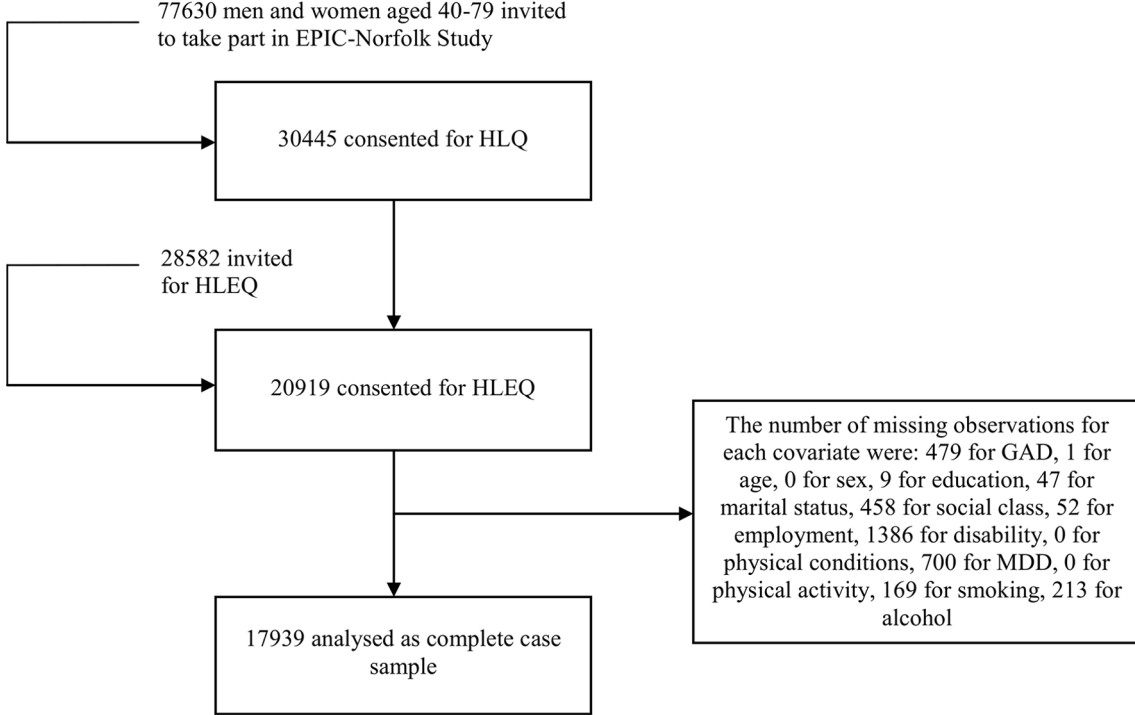

77630 men and women aged 40-79 invited to take part in EPIC-Norfolk Study

30445 consented for HLQ

28582 invited for HLEQ

20919 consented for HLEQ

The number of missing observations for each covariate were: 479 for GAD, 1 for age, 0 for sex, 9 for education, 47 for marital status, 458 for social class, 52 for employment, 1386 for disability, 0 for physical conditions, 700 for MDD, 0 for physical activity, 169 for smoking, 213 for alcohol

17939 analysed as complete case sample

Note: Some participants had missing observations on more than one covariate.

**Figure 1** Flow chart of the European Prospective Investigation of Cancer-Norfolk (EPIC-Norfolk) cohort. This is a flow chart showing the number of participants at each study stage: the number approached to participate in the EPIC-Norfolk study, the number enrolled at baseline who completed the HLQ, those who completed the psychosocial HLEQ, and finally, respondents with complete data on covariates. The EPIC-Norfolk study consists of middle-aged and older British people. GAD, generalised anxiety disorder; HLEQ, Health and Life Experiences Questionnaire; MDD, major depressive disorder.

participants provided data for the HLEQ; most of the missing observations were from past-year GAD (479), past-year MDD (700) and disability (1386); the remaining missing observations were generated from the other variables (figure 1). Notable findings from the missing data analysis show that people with missing GAD more often had pre-existing health conditions, high disability, MDD, low alcohol consumption and were without employment (online supplementary appendix 1).

The sample used for the analyses consisted of a total of 17939 participants. Respondents were assessed between 1999/2000 and 2009 (figure 1).

In 1996–2000, 393 out of 17939 (2.2%) people were identified as having past-year GAD. Table 1 shows the baseline characteristics of participants by GAD status.

Those with GAD were more likely to be younger, women, inactive, current smokers, low alcohol consumers, of higher educational attainment, single, of non-manual social class, without employment, with physical conditions, high levels of disability and MDD. Table 2 summarises the means and SDs of the number of hospital admissions by participant characteristics.

Participants with GAD had a higher frequency of hospitalisation compared with those without GAD. Some of the findings show that frequency of hospitalisation was markedly higher among older age groups, men, those with low

educational attainment, unemployed participants, those with high levels of disability and with past-year MDD.

Table 3 shows the unadjusted and adjusted incidence rate ratios (IRRs) of hospital admissions by GAD status.

After adjustment for sociodemographic variables, physical conditions and disability, GAD was associated with a 25% higher incidence rate of hospitalisation (IRR=1.25, 95% CI 1.09 to 1.43). The IRR was somewhat attenuated and became statistically non-significant after further adjustment for MDD (IRR=1.10, 95% CI 0.96 to 1.27). The effect estimate approached the null after additional adjustment for behaviour risk factors (IRR=1.04, 95% CI 0.90 to 1.20).

Next, we assessed whether risk for hospital admissions varied by frequency of GAD lifetime episodes, anxiety episode chronicity, GAD age of onset and whether the hospitalisation risk was higher in those with psychiatric comorbidity (with MDD) (table 4). Results are based on fully adjusted models.

People with >3 lifetime episodes had a somewhat higher risk of hospitalisation (IRR=1.07, 95% CI 0.91 to 1.27). Those whose episodes lasted, on average, 6 months or longer also had a slight increased risk for admissions compared with those with shorter episodes (IRR=1.07, 95% CI 0.85 to 1.35). People who developed GAD before 30 years of age were 16% more likely to be admitted to

**Table 1** Percentage and number of people with past-year generalised anxiety disorder (GAD) reported in 1996–2000 according to sociodemographic factors, health status and behaviour risk factors for the European Prospective Investigation of Cancer-Norfolk cohort (n=17 939)

| Characteristic | Number with characteristic | Percentage and number with past-year GAD |
|---|---|---|
| Sociodemographics | | |
| Age (years) | | |
| <50 | 2359 | 3.3 (79)*** |
| 50–60 | 6209 | 2.9 (179) |
| 60–70 | 5733 | 1.6 (94) |
| 70+ | 3638 | 1.1 (41) |
| Sex | | |
| Women | 9937 | 2.5 (249)* |
| Men | 8002 | 1.8 (144) |
| Education† | | |
| Low | 6106 | 2.0 (120)* |
| High | 11 833 | 2.3 (273) |
| Marital status | | |
| Single | 686 | 3.6 (25)*** |
| Married | 14 538 | 2.0 (284) |
| Other‡ | 2715 | 3.1 (84) |
| Social class | | |
| Manual | 6836 | 2.0 (137) |
| Non-manual | 11 103 | 2.3 (256) |
| Employment | | |
| Yes | 7712 | 2.0 (155) |
| No | 10 227 | 2.3 (238) |
| Health status | | |
| Physical conditions§ | | |
| Yes | 9166 | 2.7 (251)*** |
| No | 8773 | 1.6 (142) |
| Disability level | | |
| High¶ | 8900 | 3.0 (266)*** |
| Low | 9039 | 1.4 (127) |
| Psychiatric conditions | | |
| Past-year major depressive disorder | | |
| Yes | 934 | 21.4 (200)*** |
| No | 17 005 | 1.1 (193) |
| Behaviour risk factors | | |
| Physical activity | | |
| Active†† | 12 822 | 2.1 (272) |
| Inactive | 5117 | 2.4 (121) |
| Smoking status | | |
| Current smoker | 1893 | 4.7 (89)*** |
| Former smoker | 7470 | 1.9 (141) |

Continued

**Table 1** Continued

| Characteristic | Number with characteristic | Percentage and number with past-year GAD |
|---|---|---|
| Never-smoker | 8576 | 1.9 (163) |
| Alcohol intake | | |
| High‡‡ | 9241 | 2.0 (182)* |
| Low | 8698 | 2.4 (211) |

*P<0.05, ***P<0.001.
†High education: O-level, A-level, degree; low education: refers to no education.
‡Other: divorced, separated, widowed.
§Physical conditions: respiratory disease (asthma and bronchitis), allergies and hay fever, stroke, heart attack, cancer, diabetes, thyroid conditions, arthritis.
¶Below the physical component summary score value of 50.6.
††Moderately inactive, moderately active, active.
‡‡3+units of alcohol/week (one pint beer=two units, one glass wine=one unit, one glass sherry=one unit, one glass spirit=one unit).

the hospital than those who developed it later in life (IRR=1.16, 95% CI 0.95 to 1.41). Finally, we determined whether GAD comorbid with MDD is associated with hospital admissions. Results showed that people with GAD-MDD comorbidity had a 23% higher chance of being admitted to hospital than people without comorbidity—this association was statistically significant (IRR=1.23, 95% CI 1.02 to 1.49).

## DISCUSSION

This is the first study to assess the association between GAD and hospital service use in a population-based cohort. This longitudinal study showed that having an episode of GAD in the past year was not independently associated with hospital admissions during the subsequent 9 years. Chronic GAD (at least 6 months), frequent GAD (at least three lifetime episodes) and anxiety with an early age of onset (before 30 years) did not show statistically significant associations with hospitalisations. In contrast, people with GAD and MDD comorbidity were at an increased risk of being admitted to hospital than those without comorbidity with MDD. The association between GAD-MDD comorbidity and hospital admissions was statistically significant.

People with past-year GAD were more likely to have medical conditions; nonetheless, including these covariates in the model left the association between past-year GAD and hospital admissions statistically significant. It was only when MDD was introduced in the model as a potential confounder that any remaining association with hospital service utilisation was explained away.

## Strengths and limitations

There are several strengths associated with our study. We had a large, population-based sample of British adults over the age of 40, and took into account relevant confounders. We used a structured questionnaire to

**Table 2** Hospital admissions (mean, SD) by participant characteristics in 17 939 British people between 1999/2000 and 2009

| Characteristic | Total number with characteristic | Number of admissions Mean (SD) |
|---|---|---|
| Past-year generalised anxiety disorder | | |
| Yes | 393 | 4.0 (6.3)*** |
| No | 17 546 | 3.4 (13.0) |
| Sociodemographics | | |
| Age (years) | | |
| <50 | 2359 | 1.9 (9.8)*** |
| 50–60 | 6209 | 3.0 (16.5) |
| 60–70 | 5733 | 3.8 (11.2) |
| 70+ | 3638 | 4.6 (9.6) |
| Sex | | |
| Women | 9937 | 3.1 (14.0)*** |
| Men | 8002 | 3.9 (11.3) |
| Education† | | |
| Low | 6106 | 4.1 (17.1)*** |
| High | 11 833 | 3.1 (10.1) |
| Marital status | | |
| Single | 686 | 3.0 (8.9)*** |
| Married | 14 538 | 3.3 (10.9) |
| Other‡ | 2715 | 4.0 (21.0) |
| Social class | | |
| Manual | 6836 | 4.0 (18.3)*** |
| Non-manual | 11 103 | 3.1 (7.8) |
| Employment | | |
| Yes | 7712 | 2.5 (9.1)*** |
| No | 10 227 | 4.1 (15.1) |
| Health status | | |
| Physical conditions§ | | |
| Yes | 9166 | 3.9 (10.4)*** |
| No | 8773 | 3.0 (15.1) |
| Disability level | | |
| High¶ | 8900 | 4.4 (16.5)*** |
| Low | 9039 | 2.5 (7.8) |
| Psychiatric conditions | | |
| Past-year major depressive disorder | | |
| Yes | 934 | 4.5 (13.6)*** |
| No | 17 005 | 3.4 (12.9) |
| Behaviour risk factors | | |
| Physical activity | | |
| Active†† | 12 822 | 3.2 (13.3)*** |
| Inactive | 5117 | 4.1 (11.7) |
| Smoking status | | |

Continued

**Table 2** Continued

| Characteristic | Total number with characteristic | Number of admissions Mean (SD) |
|---|---|---|
| Current smoker | 1893 | 4.6 (26.8)*** |
| Former smoker | 7470 | 3.8 (11.4) |
| Never-smoker | 8576 | 2.9 (8.6) |
| Alcohol intake | | |
| High‡‡ | 9241 | 3.2 (13.3)*** |
| Low | 8698 | 3.7 (12.5) |

*P<0.05, ***P<0.001.
†High education: O-level, A-level, degree; low education: refers to no education.
‡Other: divorced, separated, widowed.
§Physical conditions: respiratory disease (asthma and bronchitis), allergies and hay fever, stroke, heart attack, cancer, diabetes, thyroid conditions, arthritis.
¶Below the physical component summary score value of 50.6.
††Moderately inactive, moderately active, active.
‡‡3+units of alcohol/week (one pint beer=two units, one glass wine=one unit, one glass sherry=one unit, one glass spirit=one unit).

assess presence of GAD in the past year according to valid and reliable criteria, used large administrative health databases to examine hospital service use (avoiding the self-reporting bias found in questionnaire studies) and participants were followed for a long time. Medical histories were derived according to a large list of self-reported physician diagnoses of chronic diseases. Despite this, some illnesses may have been missed. It is also possible that participants may have underreported certain conditions, which may have introduced measurement error. A negligible proportion of respondents may have obtained care at private facilities, and thus their health service use would not have been recorded for this study. The databases used also did not capture admissions to hospitals outside the UK. However, migration in the EPIC-Norfolk cohort is minimal and does not present a problem.

Self-evaluated impairment was also included in models as a confounder; however, this may have led to overadjustment in analyses - impairment may be part of the expression of psychiatric illness. This may have resulted in attenuation of effect estimates.

Another limitation is that we did not have data on primary care service use. Merging population cohorts, such as ours, with primary care service administrative databases and hospitalisation databases would have provided a more complete picture of the burden of GAD on the healthcare system.

This study was conducted on people ages 40 years and older and may not be generalisable to younger age groups. We suspect that the strength of the association between GAD-MDD comorbidity and hospital admissions is weaker for younger populations who are typically healthier than older people. Although young people have a high burden of mental health problems,[39 43] they

**Table 3** Associations between past-year generalised anxiety disorder (GAD) reported in 1996–2000 and hospital admissions in 1999/2000–2009 in 17 939 British people over the age of 40

| | IRR and 95% CI | | | |
|---|---|---|---|---|
| Characteristic | Crude IRR | A* | B† | C‡ |
| **Past-year GAD** | | | | |
| Yes | 1.18 (1.02 to 1.36) | 1.25 (1.09 to 1.43) | 1.10 (0.96 to 1.27) | 1.04 (0.90 to 1.20) |
| No | 1.00 | 1.00 | 1.00 | 1.00 |
| **Sociodemographics** | | | | |
| **Age** | | | | |
| Per 10 years | 1.36 (1.33 to 1.40) | 1.19 (1.16 to 1.23) | 1.20 (1.17 to 1.24) | 1.21 (1.18 to 1.25) |
| **Sex** | | | | |
| Women | 0.80 (0.76 to 0.83) | 0.76 (0.73 to 0.79) | 0.76 (0.72 to 0.79) | 0.78 (0.74 to 0.81) |
| Men | 1.00 | 1.00 | 1.00 | 1.00 |
| **Education§** | | | | |
| Low | 1.30 (1.24 to 1.36) | 1.13 (1.08 to 1.18) | 1.13 (1.08 to 1.19) | 1.11 (1.06 to 1.16) |
| High | 1.00 | 1.00 | 1.00 | 1.00 |
| **Marital status** | | | | |
| Single | 0.88 (0.79 to 0.99) | 0.85 (0.77 to 0.95) | 0.85 (0.76 to 0.95) | 0.84 (0.76 to 0.94) |
| Married | 1.00 | 1.00 | 1.00 | 1.00 |
| Other¶ | 1.21 (1.14 to 1.28) | 1.17 (1.11 to 1.24) | 1.14 (1.07 to 1.21) | 1.09 (1.03 to 1.16) |
| **Social class** | | | | |
| Manual | 1.29 (1.23 to 1.34) | 1.24 (1.19 to 1.30) | 1.24 (1.19 to 1.30) | 1.21 (1.16 to 1.26) |
| Non-manual | 1.00 | 1.00 | 1.00 | 1.00 |
| **Employment** | | | | |
| Yes | 1.00 | 1.00 | 1.00 | 1.00 |
| No | 1.64 (1.57 to 1.71) | 1.18 (1.12 to 1.25) | 1.18 (1.12 to 1.24) | 1.15 (1.09 to 1.21) |
| **Health status** | | | | |
| **Physical conditions**** | | | | |
| Yes | 1.32 (1.26 to 1.37) | 1.18 (1.13 to 1.23) | 1.17 (1.12 to 1.22) | 1.18 (1.13 to 1.23) |
| No | 1.00 | 1.00 | 1.00 | 1.00 |
| **Disability level** | | | | |
| High†† | 1.78 (1.71 to 1.86) | 1.52 (1.45 to 1.59) | 1.51 (1.44 to 1.57) | 1.48 (1.42 to 1.55) |
| Low | 1.00 | 1.00 | 1.00 | 1.00 |
| **Psychiatric conditions** | | | | |
| **Past-year MDD** | | | | |
| Yes | 1.34 (1.22 to 1.48) | | 1.34 (1.22 to 1.48) | 1.33 (1.21 to 1.46) |
| No | 1.00 | | 1.00 | 1.00 |
| **Lifestyle** | | | | |
| **Physical activity** | | | | |
| Active‡‡ | 1.00 | | | 1.00 |
| Inactive | 1.27 (1.21 to 1.33) | | | 1.04 (1.00 to 1.09) |
| **Smoking status** | | | | |
| Current smoker | 1.60 (1.49 to 1.72) | | | 1.51 (1.41 to 1.62) |
| Former smoker | 1.33 (1.27 to 1.39) | | | 1.13 (1.08 to 1.18) |
| Never-smoker | 1.00 | | | 1.00 |

**Table 3** Continued

| Characteristic | IRR and 95% CI | | | |
| --- | --- | --- | --- | --- |
| | Crude IRR | A* | B† | C‡ |
| Alcohol intake | | | | |
| High§§ | 0.88 (0.85 to 0.92) | | | 0.92 (0.88 to 0.96) |
| Low | 1.00 | | | 1.00 |

*Model A: adjusted for sociodemographics (age, sex, education, marital status, social class, employment), physical conditions, disability.
†Model B: adjusted for sociodemographics, physical conditions, disability, MDD.
‡Model C: adjusted for sociodemographics, physical conditions, disability, MDD, physical activity, smoking, alcohol.
§High education: O-level, A-level, degree; low education: refers to no education.
¶Other: divorced, separated, widowed.
**Physical conditions: respiratory disease (asthma and bronchitis), allergies and hay fever, stroke, heart attack, cancer, diabetes, thyroid conditions, arthritis.
††Below the physical component summary score value of 50.6.
‡‡Moderately inactive, moderately active, active.
§§3+units of alcohol/week (one pint beer=two units, one glass wine=one unit, one glass sherry=one unit, one glass spirits=one unit).
IRR, incidence rate ratio; MDD, major depressive disorder.

(especially adolescents) are less likely to have hospitalisations than older people.[44] It could take many years until the effects of anxiety comorbid with depression accumulate and manifest as poor physical health, thus translating into higher use of hospital services. As such, we would expect the strength of the association between GAD-MDD comorbidity and hospitalisations to be weaker in young people; however, future studies should investigate this.

In order to participate in EPIC-Norfolk, participants were required to complete extensive health and lifestyle questionnaires and undergo regular health examinations. Those who agreed to take part in the study were healthier and more affluent than people living in other regions of England; as such, findings may not generalise to individuals living in extreme deprivation.

### Comparison with other studies
Many of the studies assessing the link between psychiatric disorders and health service utilisation have focused on depression and, to a lesser extent, panic disorder and post-traumatic stress disorder (PTSD), while other anxiety disorders have been significantly under-researched. Studies on depression as a stand-alone measure have shown an association with health service use in both clinical and community samples.[45] There are substantially fewer studies on anxiety, and a number of these have shown positive associations with health service use. A US study[46] that recruited patients from an outpatient setting showed that anxiety disorders were linked to utilisation of outpatient primary and specialty medical care services. Patients, however, were recruited from a clinical setting located in a predominantly suburban/rural area, which might have affected generalisability. Another study showed anxiety disorders to be associated with a high number of consultations with general and specialty medical providers, such as those working

in cardiology and dermatology.[47] In this study, people were sampled from an anxiety clinic, thereby leading to possible selection bias. Other studies showed PTSD and GAD to be associated with healthcare use; however, this research was based on highly select samples that have limited generalisability.[25 27 48 49] In contrast to the literature, a major strength of our study was that it was population-based. There is also a lack of research assessing whether different forms of the disorder contribute to even higher health service use rates (comorbid cases are typically the most severe, hardest to treat and with the poorest prognosis[9]).

### Mechanisms
A more severe course of GAD can lead to higher rates of health services, because of unhealthy behaviours, such as smoking and alcohol (which we controlled for in our analyses). It could also be that a more severe form of anxiety, such as GAD-MDD comorbidity, is associated with poorer underlying health, which then leads to higher health service use rates. Although we adjusted for a number of chronic illnesses, it is possible that some physician-diagnosed conditions associated with GAD-MDD comorbidity and hospitalisations were not captured by the HLQ. A third explanation for higher health service use in those with comorbid anxiety and depression could relate to undiagnosed medical illness which is at an early, undetectable stage. Early symptoms of disease could be prompting people with psychiatric comorbidity to make greater use of health services.

### Implications
GAD can increase the risk for disability and impairment.[9] There is insufficient research on its association with health services, and the studies that have been conducted are small and based on clinical samples. Our study overcomes

**Table 4** Associations between different forms of generalised anxiety disorder (GAD) reported in 1996–2000 and hospital admissions in 1999/2000–2009 in 17 939 British people over the age of 40

| Characteristic | IRR and 95% CI | | | |
|---|---|---|---|---|
| **GAD type** | | | | |
| Frequent GAD | | | | |
| Yes* | 1.07 (0.91 to 1.27)† | | | |
| No | 1.00 | | | |
| Chronic GAD | | | | |
| Yes‡ | | 1.07 (0.85 to 1.35)† | | |
| No | | 1.00 | | |
| Early age GAD onset | | | | |
| Yes§ | | | 1.16 (0.95 to 1.41)† | |
| No | | | 1.00 | |
| Comorbid GAD | | | | 1.23 (1.02 to 1.49)† |
| Yes¶ | | | | 1.00 |
| No | | | | |
| **Sociodemographics** | | | | |
| Age | | | | |
| Per 10 years | 1.22 (1.18 to 1.25) | 1.21 (1.18 to 1.25) | 1.22 (1.18 to 1.25) | 1.21 (1.17 to 1.25) |
| Sex | | | | |
| Women | 0.78 (0.74 to 0.81) | 0.78 (0.74 to 0.81) | 0.78 (0.74 to 0.81) | 0.78 (0.75 to 0.82) |
| Men | 1.00 | 1.00 | 1.00 | 1.00 |
| Education** | | | | |
| Low | 1.10 (1.06 to 1.16) | 1.11 (1.06 to 1.16) | 1.10 (1.06 to 1.16) | 1.12 (1.07 to 1.18) |
| High | 1.00 | 1.00 | 1.00 | 1.00 |
| Marital status | | | | |
| Single | 0.84 (0.76 to 0.94) | 0.84 (0.76 to 0.94) | 0.84 (0.76 to 0.94) | 0.83 (0.74 to 0.93) |
| Married | 1.00 | 1.00 | 1.00 | 1.00 |
| Other†† | 1.10 (1.03 to 1.16) | 1.09 (1.03 to 1.16) | 1.09 (1.03 to 1.16) | 1.03 (0.97 to 1.10) |
| Social class | | | | |
| Manual | 1.21 (1.16 to 1.26) | 1.21 (1.16 to 1.26) | 1.21 (1.16 to 1.26) | 1.21 (1.16 to 1.27) |
| Non-manual | 1.00 | 1.00 | 1.00 | 1.00 |
| Employment | | | | |
| Yes | 1.00 | 1.00 | 1.00 | 1.00 |
| No | 1.15 (1.09 to 1.21) | 1.15 (1.09 to 1.21) | 1.15 (1.09 to 1.21) | 1.17 (1.11 to 1.24) |
| **Health status** | | | | |
| Physical conditions‡‡ | | | | |
| Yes | 1.17 (1.13 to 1.23) | 1.17 (1.13 to 1.23) | 1.17 (1.13 to 1.23) | 1.17 (1.12 to 1.22) |
| No | 1.00 | 1.00 | 1.00 | 1.00 |
| Disability level | | | | |
| High§§ | 1.48 (1.42 to 1.55) | 1.48 (1.42 to 1.55) | 1.48 (1.42 to 1.55) | 1.48 (1.42 to 1.55) |
| Low | 1.00 | 1.00 | 1.00 | 1.00 |
| Psychiatric conditions | | | | |
| Past-year MDD | | | | |
| Yes | 1.32 (1.20 to 1.45) | 1.33 (1.22 to 1.46) | 1.33 (1.21 to 1.46) | – |
| No | 1.00 | 1.00 | 1.00 | |
| Lifestyle | | | | |

Continued

| Table 4 Continued | | | | |
|---|---|---|---|---|
| **Characteristic** | **IRR and 95% CI** | | | |
| Physical activity | | | | |
| Active¶¶ | 1.00 | 1.00 | 1.00 | 1.02 (0.97 to 1.07) |
| Inactive | 1.04 (1.00 to 1.09) | 1.04 (1.00 to 1.09) | 1.04 (1.00 to 1.09) | 1.00 |
| Smoking status | | | | |
| Current smoker | 1.51 (1.41 to 1.62) | 1.51 (1.41 to 1.62) | 1.51 (1.41 to 1.62) | 1.56 (1.45 to 1.68) |
| Former smoker | 1.13 (1.08 to 1.18) | 1.13 (1.08 to 1.18) | 1.13 (1.08 to 1.18) | 1.14 (1.09 to 1.19) |
| Never-smoker | 1.00 | 1.00 | 1.00 | 1.00 |
| Alcohol intake | | | | |
| High*** | 0.92 (0.88 to 0.96) | 0.92 (0.88 to 0.96) | 0.92 (0.88 to 0.96) | 0.93 (0.89 to 0.98) |
| Low | 1.00 | 1.00 | 1.00 | 1.00 |

*3+ episodes of lifetime GAD.
†Adjusted for sociodemographics, physical conditions, disability, MDD, physical activity, smoking, alcohol.
‡GAD episodes lasted at least 6 months.
§GAD developed before 30 years of age.
¶GAD-MDD comorbidity.
**High education: O-level, A-level, degree; low education: refers to no education.
††Other: divorced, separated, widowed.
‡‡Physical conditions: respiratory disease (asthma and bronchitis), allergies and hay fever, stroke, heart attack, cancer, diabetes, thyroid conditions, arthritis.
§§Below the physical component summary score value of 50.6.
¶¶Moderately inactive, moderately active, active.
***3+ units of alcohol/week (one pint beer=two units, one glass wine=one unit, one glass sherry=one unit, one glass spirit=one unit).
IRR, incidence rate ratio; MDD, major depressive disorder.

many limitations of previous literature and clarifies that individual episodes of GAD measured at a single point in time (eg, in the past year) are not associated with health service use. Instead, our analyses show that cases that are comorbid with depression can lead to increased use of hospital services, after controlling for a range of important confounders. In this study, GAD-MDD comorbidity was associated with a statistically significantly increased risk of hospital admissions.

There is insufficient population-based research on anxiety, and studies on the association between GAD and hospitalisation admissions are lacking. Clinicians should consider that it may not necessarily be the diagnosis of the individual disorder at one point in time (eg, past-year GAD) that is predictive of deleterious health outcomes; different forms of the disorder may be more important. GAD has a waxing and waning course throughout a patient's life, and many of those affected experience relapse after psychiatric treatment or develop psychiatric comorbidities.

Our findings are important for clinicians and policy-makers. A number of people are affected by anxiety–depression comorbidity.[9] As such, clinicians should consider more widespread screening for mental health problems, and if appropriate, the examination of any underlying health conditions that may require treatment in order to prevent future hospital admissions. Policymakers should also consider supporting the widespread roll-out of anxiety and depression prevention and screening programmes.

Future research, however, needs to examine the reasons for the increased hospital service use in those with GAD-MDD comorbidity (this can provide additional insight into clinical and policy recommendations). To provide a better understanding of the links between mental and physical health, the bidirectional links between anxiety and physical health problems should also be examined. Finally, future research should merge a population-based cohort with primary and secondary care administrative health databases to provide a more complete picture of the burden of different forms of anxiety on the healthcare system.

### CONCLUSION
People with GAD that was comorbid with MDD had a higher risk for hospital admissions in the longitudinal EPIC-Norfolk study.

**Twitter** @oliviaroxann

**Contributors** OR had the idea for and conducted the analysis, and wrote the article. CB critically reviewed drafts of the manuscript. K-TK edited versions of the paper. PS and NW provided feedback into the analysis. All authors contributed to the interpretation of data for the work, agreed to be accountable for all aspects of the work, gave final approval of the version to be published and made substantial contributions to the analysis and interpretation of data. All authors have seen and approved the final version. All authors had full access to all the data in the study and take responsibility for the integrity of the data and the accuracy of the data analysis. OR acts as guarantor of the study.

**Funding** This work was supported by the Medical Research Council UK (grant number SP2024-0201 and SP2024-0204) and Cancer Research UK (grant number G9502233).

**Competing interests**  OR received a PhD studentship from the National Institute for Health Research.

**Patient consent**  Obtained.

**Ethics approval**  Norfolk Ethics Committee (Rec Ref: 98CN01).

**Provenance and peer review**  Not commissioned; externally peer reviewed.

**Data sharing statement**  No additional data available.

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
