## [Reviewer comments · BMJ Open]

ARTICLE DETAILS

TITLE (PROVISIONAL)	Generalized anxiety disorder and non-psychiatric hospital admissions: findings from a large, population cohort study
AUTHORS	Remes, Olivia; Wainwright, Nicholas; Surtees, Paul; LaFortune, Louise; Khaw, Kay-Tee; Brayne, Carol

VERSION 1 – REVIEW

REVIEWER	Mark E. Kunik, MD, MPH Baylor College of Medicine, USA
REVIEW RETURNED	04-Aug-2017

GENERAL COMMENTS	This well-done longitudinal, population cohort designed study examines the association between generalized anxiety disorder (GAD) and hospital service use. The strengths of the study include a large population cohort, socio-demographic characterization of sample, validated measures of GAD and Major Depressive disorder (MDD), statistical approach and balanced discussion of findings and limitations. The major limitation is the study question and findings are quite narrow with questionable theoretical or clinical import. 1. The only health service measure evaluated is hospital use. Although this certainly is a marker for health status, its importance or meaning as a stand alone is not known. The authors do a nice job of discussing this as a limitation.2. I question the justification for running models that adjust for medications that are often used in the treatment of GAD. What is the rationale for this? Are there social science references or theoretical models that support this.3. Although not as critical as “#2” I also wonder about the generalizability and clinical meaning of the findings given the decision to adjust for MDD. As the author’s correctly point out, GAD and MDD are often co-morbid, with some studies showing this co-morbid group to have worse prognosis and outcomes. Adjusting for MDD essentially biases the sample to those that are likely less ill. Although this is apparent across the three models, the authors conclusions are based largely on models that adjust for MDD. In addition, as the authors point out primary care providers do not recognize GAD and they are certainly are not recognizing the difference between GAD with or without MDD. In fact, I would guess they are more likely to recognize the comorbid group.4. Given the competing demands, limited time, and unknown import of early onset-increased hospital use finding, I think last line of conclusion is not a reasonable ask of primary care providers.
--

REVIEWER	Dr. Irene Bobevski Monash University, Australia
REVIEW RETURNED	08-Aug-2017

GENERAL COMMENTS	This manuscript examines the contribution of generalized anxiety disorder (GAD) to non-psychiatric hospitalisations in a large British population survey of people aged 40 years and over. The study found that GAD was not independently associated with hospital admissions, although participants with GAD onset before 30 years of age were at increased risk of hospitalisation. The paper is clearly written and the topic is interesting and contributes to knowledge, as anxiety disorders present a considerable burden in the population. Abstract p. 2, l.39: It would make the objective of the study clearer if hospital admissions are referred to as non-psychiatric in the abstract. Introduction p.4, last sentence of 1st paragraph: Evidence that anxiety disorders are under-recognised and mismanaged by clinicians needs to be cited. p.4, l.106-108: Which specific studies have shown patients with mental health problems present with physical, rather than psychiatric symptoms? A reference is given to an edited book, but it is not clear which studies have actually found this. p.4, l.120-124: Physical illness can also lead to or exacerbate anxiety. This bi-directional relationship needs to be considered in the introduction and discussion. p.5., l.127: The CHD abbreviation seems to appear here first without being spelled out previously. p.5, l.147-150: Rather than referring to hospital service use and "higher rate of hospital services" (l.149-150) it would be more appropriate to refer to non-psychiatric hospital admissions, as this is what the paper is focusing on. Methods p.6, l.166: Is there a reference for the HLEQ? p.7.,l.199-204: Was the male partner's occupation used to assign social class to women participants? Why was not women's own occupation used where this information was available? Statistical Analysis p.9, l. 271-277: Was the analysis on risk of hospitalisation by onset and duration of GAD episodes conducted on the whole sample or only on participants with GAD? This needs to be clarified. In this sections it says that two-sided statistical tests were conducted for this analysis - what tests were they? However, in the Results section on p.20, IRRs are cited. Are the IRRs adjusted or unadjusted? The authors need to explain what was done. It would also be useful to include this analysis in a table together with any variables that were adjusted for, since this is the main significant finding of the paper.
---

Results

As a large number of participants were excluded because of missing data, missing data analysis needs to be reported. In what way did those with missing data differ from those retained in the analysis? Does analysis with all participants, using all the available data (or at least where GAD status is available), differ from the analysis with all missing data excluded? Should imputations be considered for any variables with missing data?

p.14 and p.17: It would be useful to test which of these differences are statistically significant.

Tables

In the tables it would be more appropriate to refer to physical and psychiatric conditions, rather than "comorbidities", as the tables refer to the whole sample and these may not necessarily be comorbid for everyone.

Table 1, p.13 and Table 2, p.16: There is an extra column that does not make sense.

Discussion

p. 21: lines 391-393 are repetitive of lines 381-382.

Limitations

The self-report measure of physical conditions needs to be acknowledged as a limitation. A large amount of missing data is also a limitation and some more detailed discussion of this is required, based on missing data analysis.

Another limitation that is not acknowledged is that only people aged 40 years or over were included in the study, and it is not known to what extent the results can be generalised to younger people. Perhaps relevant literature on age differences in service use and hospitalisations could be cited.=

VERSION 1 – AUTHOR RESPONSE

Reviewer 1

Reviewer Name: Mark E. Kunik, MD, MPH

Institution and Country: Baylor College of Medicine, USA Please state any competing interests or state 'None declared': None declared

Comments and Responses:

1. This well-done longitudinal, population cohort designed study examines the association between generalized anxiety disorder (GAD) and hospital service use. The strengths of the study include a large population cohort, socio-demographic characterization of sample, validated measures of GAD and Major Depressive disorder (MDD), statistical approach and balanced discussion of findings and limitations. The major limitation is the study question and findings are quite narrow with questionable theoretical or clinical import.

Response: Thank you very much for the feedback. Thank you for mentioning that this study is well-done, that this research has strengths, and that we provided a balanced discussion. My co-authors and I have made all the changes requested. We included an analysis on GAD-MDD comorbidity, and we agree that now, it substantially strengthens this paper.

2. The only health service measure evaluated is hospital use. Although this certainly is a marker for health status, its importance or meaning as a stand alone is not known. The authors do a nice job of discussing this as a limitation.

Response: Thank you very much.

3. I question the justification for running models that adjust for medications that are often used in the treatment of GAD. What is the rationale for this? Are there social science references or theoretical models that support this.

Response: We agree. As per your statement and in accordance with other publications that have assessed the link between mental disorders and health service use (ex. Hamalainen 2008), we have omitted medications that are used in the treatment of GAD from the models.

4. Although not as critical as “#3” I also wonder about the generalizability and clinical meaning of the findings given the decision to adjust for MDD. As the author’s correctly point out, GAD and MDD are often co-morbid, with some studies showing this co-morbid group to have worse prognosis and outcomes. Adjusting for MDD essentially biases the sample to those that are likely less ill. Although this is apparent across the three models, the authors conclusions are based largely on models that adjust for MDD. In addition, as the authors point out primary care providers do not recognize GAD and they are certainly are not recognizing the difference between GAD with or without MDD. In fact, I would guess they are more likely to recognize the comorbid group.

Response: Thank you very much for this comment. We agree that GAD-MDD comorbidity is more severe and associated with worse outcomes than either disorder alone. Also, a high number of people are affected by both anxiety and depression. As such, we have included a model with GAD-MDD comorbidity, and updated the Introduction, Methods, Results, and Discussion sections to reflect this. We included results from this analysis in table 4.

Recent evidence (ECNP 2016) also suggests that anxiety is often the primary disorder and can increase the risk for depression – thus, as per your comment, people with both anxiety and depression constitute a sicker group and would most likely be the highest consumers of health services (compared to people with either pure anxiety or pure depression). However, because of word count restrictions, we were unable to comment on this or on the fact that primary care providers may have difficulty recognizing pure disorders.

Just an additional note - reviewer 2 indicated that the association we focused on (anxiety and health service use) is important; therefore, we left the first analysis in the paper (link between past-year GAD and health service use while adjusting for covariates including MDD). The literature on the link between anxiety and health service use is conflicting and has many limitations. We felt it was important to clarify this using a large, population-based study and show that anxiety measured at one point in time (ex. GAD in the past year) is not predictive of deleterious health outcomes. Rather, it may be that different forms of anxiety, such as GAD with an early age of onset and especially GAD that is comorbid with MDD, are important.

5. Given the competing demands, limited time, and unknown import of early onset-increased hospital use finding, I think last line of conclusion is not a reasonable ask of primary care providers.

Response: As requested, we have now deleted the last line and changed the concluding statement.

Reviewer: 2

Reviewer Name: Dr. Irene Bobevski

Institution and Country: Monash University, Australia Please state any competing interests or state 'None declared': None declared

Please leave your comments for the authors below Review of manuscript bmjopen-2017-018539: Generalized anxiety disorder and health service use: findings from a large, population study

1. This manuscript examines the contribution of generalized anxiety disorder (GAD) to non-psychiatric hospitalisations in a large British population survey of people aged 40 years and over. The study found that GAD was not independently associated with hospital admissions, although participants with GAD onset before 30 years of age were at increased risk of hospitalisation.

The paper is clearly written and the topic is interesting and contributes to knowledge, as anxiety disorders present a considerable burden in the population.

Response: I would like to thank you for the comments you sent me. They have substantially improved this paper and we have revised several sections in accordance with the feedback you provided. Thank you for your comments and for taking the time to go over our analyses and text. My co-authors and I made almost all requested changes. We included an additional table on GAD frequency, chronicity, and age of onset; we conducted multiple imputations for missing data; clarified the text and provided relevant references where needed, and made other revisions, as suggested.

As per reviewer 1's comments, we also included an additional analysis on GAD that is comorbid with MDD.

Abstract

2. p. 2, l.39: It would make the objective of the study clearer if hospital admissions are referred to as non-psychiatric in the abstract.

Response: We have now added 'non-psychiatric' both to the abstract and other sections of the paper, and also changed the title to reflect this. We agree that this change will clarify to the reader that we are referring to non-psychiatric hospital admissions.

Introduction

3. p.4, last sentence of 1st paragraph: Evidence that anxiety disorders are under-recognised and mismanaged by clinicians needs to be cited.

Response: We have now referenced this statement.

4. p.4, l.106-108: Which specific studies have shown patients with mental health problems present with physical, rather than psychiatric symptoms? A reference is given to an edited book, but it is not clear which studies have actually found this.

Response: Initially, we had expanded on and clarified this section (and referenced relevant studies). However, after making the other corrections you and reviewer 1 had suggested, we realized that we had greatly exceeded the word limit. Thus, we felt that we could omit this small section (as it did not pertain to our objectives) and still make the arguments we wanted to make and cite the relevant literature to support our paper.

5. p.4, l.120-124: Physical illness can also lead to or exacerbate anxiety. This bi-directional relationship needs to be considered in the introduction and discussion.

Response: As suggested, we have now added in a section to the Introduction to clarify this. In the previous version of the paper, we mentioned that anxiety is linked to HPA-axis dysregulation and inflammation, and this can lead to poor health. We indicated how a recent study showed that people with anxiety disorders have more physical comorbidities, including cardiovascular diseases and their risk factors, compared to people without anxiety disorders. In this version of the manuscript, we clarify that the link between anxiety and poor health is bidirectional. It is not just anxiety that can increase the risk for poor health. Anxiety can also represent a response to an underlying medical illness.

We thus added in the following to the Introduction:

“Conversely, anxiety could also represent a response to underlying medical illness and physical illness can exacerbate anxiety; the possibility of a bidirectional relationship between anxiety and physical health should not be excluded.[14, 15] Compelling evidence from prospective studies, however, has shown that anxiety can indeed increase the risk of serious chronic conditions, such as cancer[16] and CHD[17].”

In the Discussion, we added the following sentence:

“To provide a better understanding of the links between mental and physical health, the bidirectional links between anxiety and physical health problems should also be examined.”

Unfortunately, we could not expand on our argument in the Discussion, because of word count restrictions. However, we referenced studies in support of the 'bidirectional' statement in the Introduction section and provide examples.

6. p.5., l.127: The CHD abbreviation seems to appear here first without being spelled out previously.

Response: Thank you for noting this. We made sure to spell out CHD in the first instance of mentioning it.

7. p.5, l.147-150: Rather than referring to hospital service use and "higher rate of hospital services" (l.149-150) it would be more appropriate to refer to non-psychiatric hospital admissions, as this is what the paper is focusing on.

Response: We have made the change, as requested. Throughout the paper, we also made this change in other instances as well, because we felt it was clearer to refer to non-psychiatric hospital admissions. We also changed the title of the paper to reflect this.

Methods

8. p.6, l.166: Is there a reference for the HLEQ?

Response: We have now added in the relevant reference.

9. p.7.,l.199-204: Was the male partner's occupation used to assign social class to women participants? Why was not women's own occupation used where this information was available?

Response: We have clarified the information on social class. We updated the Methods section with the following:

"For men, social class was coded using their own occupation except when they were unemployed or retired in which case their partner's social class was used. Unemployed men without partners were unclassified. Social class in women was based on their partner's except when the partner's social class was unclassified, missing, or they had no partner in which case social class was based on their own occupation. An unemployed woman without a partner was coded as unclassified."

Hence, if the woman had no partner or her partner's social class information was unavailable, then the woman's social class was based on her own occupation. Back when this study was started (1993-1997), the decision was made to assign women's social class according to their partner's occupation. Unfortunately, we have no other information on social class available to us. Nevertheless, all publications that have arisen from the EPIC-Norfolk study have used this measure of social class.

Statistical Analysis

10. p.9, l. 271-277: Was the analysis on risk of hospitalisation by onset and duration of GAD episodes conducted on the whole sample or only on participants with GAD? This needs to be clarified.

Response: The analysis was conducted on the whole sample. We have now clarified the following section:

"Finally, we determined whether the risk for hospitalization was higher among those with: 1) 3 or more episodes of lifetime GAD (versus those with fewer than 3 episodes or no GAD), 2) episodes that lasted on average 6 months or more (versus those with fewer than 6 months or no GAD), 3) age of onset at 30 years or younger (versus people with age at onset over 30 years or no GAD), and 4) psychiatric comorbidity with MDD (versus no GAD-MDD comorbidity)."

11. In this sections it says that two-sided statistical tests were conducted for this analysis - what tests were they? However, in the Results section on p.20, IRRs are cited. Are the IRRs adjusted or unadjusted? The authors need to explain what was done. It would also be useful to include this analysis in a table together with any variables that were adjusted for, since this is the main significant finding of the paper.

Response: When we conducted multivariate models, SAS produced p-values corresponding to each covariate/effect estimate – the p-values, which are two-sided, indicate whether the effect estimate (in this case, the incidence rate ratio) is significantly different from 1. We included the following phrase in our Methods:

“Two-sided statistical tests for the maximum likelihood zero inflation parameter estimates were conducted and a p-value of <0.05 was used for statistical significance.”

When we discuss the risk for hospital admissions according to frequency of GAD lifetime episodes, age of GAD onset, episode chronicity, and comorbidity with MDD, the IRRs are fully adjusted (we have now clarified this in the Results section and footnote of table 4). In the Results, we now state that the findings “are based on fully-adjusted models.” In the footnote of table 4, we provide the covariates that were adjusted for.

As indicated, we have also included the results for this analysis together with all the covariates that were adjusted for in a separate table (4).

Results

12. As a large number of participants were excluded because of missing data, missing data analysis needs to be reported. In what way did those with missing data differ from those retained in the analysis? Does analysis with all participants, using all the available data (or at least where GAD status is available), differ from the analysis with all missing data excluded? Should imputations be considered for any variables with missing data?

Response: Thank you for this comment - as requested, we have made this change. We also created 2 appendices containing further information.

First, we determined the number of people with missing GAD (our exposure variable) for each level of the covariate/for each characteristic. We also determined whether any of the differences were statistically significant at the $p < 0.05$ level (please see Appendix I) – if they were, we indicated this in the table and reported the findings in the Results section: “Notable findings from the missing data analysis show that people with missing GAD were more likely to have pre-existing health conditions, high disability, MDD, were without employment, and were more likely to report low alcohol consumption (Appendix I).”

The table in Appendix I includes results from the initial analysis of missing data.

Next, we conducted multiple imputations for missing data. We updated the Methods, Results, and Discussion sections to reflect this. Appendix II contains details on our multiple imputations. It was reassuring to see that the effect estimate remained unchanged after conducting multiple imputations. In the body of the paper we presented the results for our main analysis and primary objective: the relationship between past-year GAD and non-psychiatric hospital admissions.

13. p.14 and p.17: It would be useful to test which of these differences are statistically significant.

Response: As requested, we have now done so. We tested all of the differences, and we marked the ones that are statistically significant at the $p < 0.001$ and $p < 0.05$ levels. The footnotes of tables 1 and 2 indicate the superscripts used to denote differences that are statistically significant at $p < 0.001$ and $p < 0.05$ levels.

The Methods section has now updated to reflect this, as well:

“First, demographics, social class, medical and psychiatric conditions, and risk behaviours were compared by GAD status - the chi-square test was used to determine whether differences were statistically significant for categorical variables. Second, the mean number of hospital admissions was determined for each characteristic/covariate - the Kruskal Wallis test was used to determine statistical significance for categorical covariates with three or more groups, while the Wilcoxon rank-sum test was used for dichotomous covariates.”

Regarding table 2 – the outcome in this study, which is number of hospital admissions, was skewed. As such, we had to use the Kruskal Wallis and Wilcoxon rank-sum tests to determine whether the differences in the mean number of hospital admissions between groups were statistically significant.

Table 1 – all covariates and GAD are categorical; thus we used the chi-square test to determine whether any differences were statistically significant.

Tables

14. In the tables it would be more appropriate to refer to physical and psychiatric conditions, rather than "comorbidities", as the tables refer to the whole sample and these may not necessarily be comorbid for everyone.

Response: We agree and have now updated the tables to include 'physical conditions' and 'psychiatric conditions' as headings. We also made this change throughout the text.

The only time when we refer to comorbidities is when we discuss the analyses on GAD-MDD comorbidity, as per reviewer 1's comments. When we mention psychiatric comorbidity or GAD-MDD comorbidity, we refer to participants diagnosed with both anxiety and depression.

15. Table 1, p.13 and Table 2, p.16: There is an extra column that does not make sense.

Response: We believe you may be referring to the numbers appearing next to the table, which are the line numbers generated by Microsoft Word. These line numbers will not be appearing in the paper if it is published.

In regards to Table 1, the first column includes the total number of people with each characteristic. For example, there are 9937 women and 8002 men in this study. The second column indicates the percentage and number of people with past-year GAD. The second column shows that there are 249 out of 9937 women (2.5%) with past-year GAD, while there are 144 out of 8002 men (1.8%) with past-year GAD. We included a similar table in another paper we had published in BMJ Open (Remes 2017).

Table 2: The first column shows the total number of participants with each characteristic. For example, there are 393 people with GAD and 17546 people without it. The second column indicates the mean (standard deviation) number of hospital admissions for each characteristic/each level of the covariate. For example, the mean number of hospital admissions for people with GAD is 4.0 (sd=6.3), while the mean number of admissions for people without GAD is 3.4 (sd=13.0).

Discussion

16. p. 21: lines 391-393 are repetitive of lines 381-382.

Response: Thank you for pointing this out. We have now removed lines 391-393 from the old version.

Limitations

17. The self-report measure of physical conditions needs to be acknowledged as a limitation. A large amount of missing data is also a limitation and some more detailed discussion of this is required, based on missing data analysis.

Response: In regards to the comment on missing data analysis:

We have now added the following sentence to the Discussion: "Finally, there was missing data in this study. When we conducted multiple imputations for missing data, the effect estimate of our main analysis remained unchanged." Also, our confidence intervals become even narrower after multiple imputations were conducted, which shows that we can have even greater confidence in our findings (greater precision). I was unable to expand further on this, because of word count restrictions. However, I included an appendix detailing the steps taken in the multiple imputation for missing data, and the Results section contains the findings for this analysis.

Thank you for the comment on the self-report measure of physical conditions. In the text, we had mentioned that we had access to a "large list of self-reported physician diagnoses of chronic diseases that we used to ascertain medical histories. Despite this, the residual effect of diseases not captured by our study, but that are associated with GAD may be present. Past illness may have been underreported, which may have introduced measurement error and further attenuated effect estimates towards the null."

Thus, we mention that (because of the self-report nature of the data collection process), "past illness may have been underreported, which may have introduced measurement error and further attenuated effect estimates towards the null." This was mentioned in the Discussion section, under Strengths and Limitations.

18. Another limitation that is not acknowledged is that only people aged 40 years or over were included in the study, and it is not known to what extent the results can be generalised to younger people. Perhaps relevant literature on age differences in service use and hospitalisations could be cited.

Response: Thank you for this comment. We have now added in the following section to the Discussion:

"This study was conducted on people ages 40 years and older and may not be generalizable to younger age groups. We suspect that the strength of the association between GAD-MDD comorbidity and non-psychiatric hospital admissions is weaker for younger populations, who are typically healthier than older people. Although young people have a high burden of mental health problems[40, 44], they (especially adolescents) are less likely to have non-psychiatric hospitalizations than older people[45]. It could take many years until the effects of anxiety comorbid with depression accumulate and manifest as poor physical health, thus translating into higher use of non-psychiatric hospital services.

As such, we would expect the strength of the association between GAD-MDD comorbidity and hospitalizations to be weaker in young people, however, future studies should investigate this.”

We indicate that younger people make less use of hospital services than older people – this is data from the UK Hospital Episode Statistics – we now referenced this source.

I would like to thank you once again for providing us with these comments – they have substantially improved this paper.

VERSION 2 – REVIEW

REVIEWER	Mark Kunik, MD, MPH Baylor College of Medicine, Houston TX USA
REVIEW RETURNED	09-Oct-2017

GENERAL COMMENTS	Authors did a find job of responding to reviews. In particular, I believe inclusion of MDD comorbidity in analyses and discussion adds to the importance of the findings.
---

REVIEWER	Dr. Irene Bobevski Monash University, Australia
REVIEW RETURNED	17-Oct-2017

GENERAL COMMENTS	The authors have addressed my comments well in the revised manuscript. However, in the revised analysis the association between early onset GAD and non-psychiatric hospital admissions is no longer statistically significant although it approaches significance (OR=1.16; CI 0.95-1.41). The authors still refer to this result throughout the manuscript as if it was significant. Please report and discuss this result more accurately.
---

VERSION 2 – AUTHOR RESPONSE

Reviewer: 1

Reviewer Name: Mark Kunik, MD, MPH :

Comment 1. Authors did a fine job of responding to reviews. In particular, I believe inclusion of MDD comorbidity in analyses and discussion adds to the importance of the findings.

Response: Thank you very much.

Reviewer: 2

Reviewer Name: Dr. Irene Bobevski

2. The authors have addressed my comments well in the revised manuscript. However, in the revised analysis the association between early onset GAD and non-psychiatric hospital admissions is no longer statistically significant although it approaches significance (OR=1.16; CI 0.95-1.41). The authors still refer to this result throughout the manuscript as if it was significant. Please report and discuss this result more accurately.

Response: Thank you for this and for indicating that we addressed the comments well.

* As requested, we made this additional change throughout the manuscript to reflect that only GAD-MDD comorbidity emerged as statistically significant, while the other forms of GAD (chronic GAD, frequent GAD, anxiety with early age of onset) did not. We have updated both the Abstract and Discussion sections to reflect this. Early onset GAD was not significantly associated with health service use - we have now clarified this.

* In line with your comment, we have also made the following small modification to the Results section – we now indicate that the association between early onset GAD and non-psychiatric hospital admissions is not statistically significant. This is how it currently reads:

“People who developed GAD before 30 years of age were 16% more likely to be admitted to the hospital than those who developed it later in life (IRR=1.16, 95% CI: 0.95, 1.41), although this finding was not statistically significant.”

* In summary, as per the suggestions received, the first paragraph of the Discussion section has been modified – we clarify to the reader that none of the forms of GAD (chronic, frequent, early age of anxiety onset) are statistically significantly associated with health service use, with the exception of comorbid GAD. The association between GAD-MDD comorbidity and health service use is the only relationship that emerged as statistically significant. The remainder of the Discussion section has been modified to follow this argument.

Thank you, once again, for reviewing our paper.